



# Gaining Hydrological Insights Through Wilk's Feature Importance: A Test-Statistic Interpretation method for Reliable and Robust Inference

Kailong Li[1], Guohe Huang[1], Brian Baetz[2]

[1]Faculty of Engineering, University of Regina, Regina, Saskatchewan, Canada S4S 0A2
[2]Department of Civil Engineering, McMaster University, Hamilton, Ontario, Canada L8S 4L8.
*Correspondence to*: Guohe Huang (huangg@uregina.ca)

**Abstract.** Feature importance has been a popular approach for machine learning models to investigate the relative significance of model predictors. In this study, we developed a Wilk's feature importance (WFI) method for hydrological inference. Compared with conventional feature
importance methods such as permutation feature importance (PFI) and mean decrease in impurity (MDI), the proposed WFI aims to provide more reliable importance scores that could partially address the equifinality problem in hydrology. To achieve this, the WFI measures the importance scores based on Wilk's $\Lambda$ (a test-statistic that can be used to distinguish the differences between two or more groups of variables) throughout a decision tree. The WFI has an advantage over PFI
and MDI as it does not account for predictive accuracy so the risk of overfitting will be greatly reduced. The proposed WFI was applied to three interconnected irrigated watersheds located in the Yellow River Basin, China. By employing the recursive feature elimination approach, our results indicated that the WFI could generate more stable relative importance scores in response to the reduction of irrelevant predictors, as compared with PFI and MDI embedded in three
different machine learning algorithms. In addition, the comparative study also shows that the predictors identified by WFI achieved the highest predictive accuracy on the testing dataset, which indicates the proposed WFI could identify more informative predictors among many irrelevant ones. We also extended the WFI to the local importance scores for reflecting the varying characteristics of a predictor in the hydrological processes. The related findings could help to gain
insights into different hydrological behaviours.

## 1 Introduction

Machine learning (ML) has been used for hydrological forecasting and examining hydrological modeling processes underpinned by statistical and physical relationships. Due to the rapid progress in data science, increased computational power, and the recent advances in ML, the predictive
accuracy of hydrological processes has been greatly improved (Reichstein et al., 2019;Shortridge



et al., 2016). Yet the descriptive power (also known as interpretability) of ML models for hydrological inference has not increased apace with their predictive power for hydrological forecasting (Konapala and Mishra, 2020). Previous studies have indicated that purely pursuing predictive accuracy may not be a sufficient reason for applying a certain hydrological model to a

given problem (Beven, 2011). Recent studies have addressed the importance of extracting interpretable information and knowledge from big data, to evolve our understanding of nature's laws behind the ML modeling processes (Murdoch et al., 2019;Reichstein et al., 2019). In hydrology, studies have indicated that the hydrological inference through ML has the potential to deal with the problem of equifinality that exists in most physically-based hydrological model

descriptions (Schmidt et al., 2020;Shortridge et al., 2016). In hydrology, equifinality means different hydrological model structures and/or parameter sets describe similar observed behaviors with similar accuracy (Beven, 2011). As a result, the same hydrological behavior can thus be described by the non-unique parameter sets associated with different physical laws. This problem could severely hamper our understanding of the underlying functioning in the hydrologic system

(Clark et al., 2011). One possible cause of this problem is that the physically-based hydrological models are constrained by additional physical information provided through a priori knowledge of hydrologic functioning encoded within both model structure and states and flux relationships (Clark et al., 2015;Schmidt et al., 2020). The ML models on another hand, are more flexible than physically-based hydrological models as they can approximate any complex relationships without

relying on additional physical information, thus structural and parameterization errors can be greatly reduced (Nearing et al., 2016). The above motivations have led us to improve the interpretability of ML models to gain reliable information for hydrological inference.

The main idea of model interpretation is to understand the model decisions, including the main

aspects of (i) identifying the most relevant predictors (i.e., independent variables) leading to model predictions and (ii) reasoning why certain predictors are responsible for a particular model response. The model interpretation for ML is mainly achieved through feature importance, which relies on techniques that quantify and rank the statistical significance of input predictors and their effect on the model response. The obtained importance scores (i.e., statistical significance of input

predictors) can be used to explain certain predictions through relevant knowledge (Scornet, 2020). Feature importance methods can be categorized as model-agnostic and model-specific (Molnar,





2020). The model-agnostic methods refer to extracting post-hoc explanations by treating the original model as a black box (Ribeiro et al., 2016b). This can be achieved by learning an interpretable model based on the outputs of the black box model (Craven and Shavlik, 1996) and

perturbing inputs, and seeing the response of the black-box model (Ribeiro et al., 2016a). Such methods mainly include permutation feature importance (PFI) (Breiman, 2001), partial dependence (PD) plots (Friedman, 2001), individual conditional expectation (ICE) plots (Goldstein et al., 2015), accumulated local effects (ALE) plots (Apley and Zhu, 2016), local interpretable model-agnostic explanations (LIME) (Ribeiro et al., 2016a) and Shapley values

(Lundberg and Lee, 2017;Shapley, 1953). In hydrology, Yang and Chui (2020) used Shapley values to explain individual predictions of hydrological response in sustainable drainage systems at fine temporal scales. Worland et al. (2019) used the LIME to infer the relation between basin characteristics and the predicted flow duration curves. Konapala and Mishra (2020) used partial dependence plots to understand the role of climate and terrestrial components in the development

of hydrological drought. Compared with the above model-agnostic methods, PFI is more widely used in hydrological inference owing to its high efficiency and its ability to take global insights into model behaviors (Molnar, 2020). Recent applications of PFI include inferring the relationship between basin characteristics and predicted low flow quantiles (Ahn, 2020) and comparing the interpretability among multiple machine learning models in the context of flood events (Schmidt

et al., 2020). The above model-agnostic methods are particularly useful for comparative studies of ML models where the underlying algorithmic structure is exceedingly complex for direct extraction of interpretable information from big data.

When compared to the model-agnostic methods, the model-specific methods (also known as

interpretable models) such as decision trees and sparse regression models, can inspect model components directly (e.g., through the paths in a decision tree or the weight of a specific predictor in a linear model) (Ribeiro et al., 2016b). In fact, regression tree ensembles (RTEs) as one of the important branches in ML, are composed of hundreds of interpretable models (i.e., decision trees). As long as the predictive performance is satisfied, a reasonable inference can be achieved through

statistical summaries (e.g., mean decrease in node impurity (Breiman, 2001) or how often a predictor has been used for node splitting (Chen et al., 2015)) of the decision trees. In hydrology, RTEs have been receiving increasing attention for hydrological forecasting owing to their superior



predictive accuracy, yet their usefulness for hydrological inference in terms of using such a transparent inference process is still limited. Such studies can be found from Worland (2018) and

Lawson et al. (2017). A possible reason causing the interpretable models to be less preferrable than model-agnostic interpretation methods is that people believe higher predictive accuracy can potentially lead to more faithful inference (Murdoch et al., 2019). Nevertheless, interpretable models such as decision trees are still considered understandable tools for inferring a particular model behavior because the transparent decision-making process functions similarly to how the

human brain makes decisions for a series of questions.

Even though both types of interpretation methods have been applied for hydrological inference, they possess several drawbacks. The model-agnostic interpretation methods assume that the same predictive accuracy will lead to the same, or at least similar inferences (i.e., importance scores).

However, Schmidt et al. (2020) disclosed that such an assumption may not be valid since the problem of equifinality (which exists in conventional hydrological model descriptions) also exists for ML model inferences (i.e., different importance scores can be observed from different ML models). Such inconsistency in the inference may hamper effective reasoning for hydrological processes. The interpretable models, on the other hand, suffer less from the equifinality problem

since the importance scores can be inspected internally (e.g., through paths of a decision tree). However, Scornet (2020) revealed that the interpretability of the mean decrease in impurity (MDI) (for measuring the importance scores of the classification and regression trees (CART) (Breiman et al., 1984)) is strongly affected by the multicollinearity of input predictors: estimated important scores are biased towards positively correlated predictors. Similar discoveries also can be found

in Strobl et al. (2007). This has been a challenge for hydrological inference since the input predictors are most likely to be correlated to each other (Robertson et al., 2013). Therefore, such bias may affect the model inference. Moreover, existing interpretable models can only provide a global importance score for each predictor, without the capacity of reflecting the effects from varied predictor characteristics (i.e., local importance scores), an issue of increasing concern for

the earth sciences communities (Reichstein et al., 2019).

Therefore, as the extension of the previous efforts, the objective of this study is to develop a Wilks feature importance (WFI) method for providing reliable and robust hydrological inference from





decision trees and RTE. Compared with model-agnostic interpretation methods, the WFI is
expected to have reduced effects of equifinality for hydrological inference since it is not related to
the model predictive accuracy. When compared with the MDI as embedded in CART, WFI is
expected to provide an unbiased estimation of importance scores owing to the advantage of the
Wilks' Λ test statistics, which thereby, could lead to improved inference robustness.

This research also entails (i) evaluation of WFI performance under the RTE framework; (ii)
comparative assessment of inference robustness, through the recursive feature elimination
approach; (iii) development of stratified Bayesian inference approach for extending the WFI to
characterize the varying roles of a predictor in the hydrological process; (iv) application of the
developed WFI to three interconnected watersheds in the Yellow River Basin.

## 2 Problem Statement

To explain the reason why the importance scores from MDI method can be a potential problem for
hydrological inference, a brief illustration of classification and regression trees (CART) (Breiman
et al., 1984) is given as follows:

The principle of CART is to successively split the training data space (i.e., predictors and response)
into many irrelevant subspaces. These subspaces along with the splitting rules will form a decision
tree, which asks each of the new observations a series of "Yes/No" questions and guides it to the
corresponding subspaces. The model prediction for a new observation shares the same value as the
average value for the training responses in that particular subspace.

The tree deduction process is illustrated using a hydrological dataset (Figure 1) including three
predictors as $X_1$ (i.e., precipitation), $X_2$ (i.e., 3-day cumulative precipitation) and $X_3$ (i.e.,
temperature), and a response $Y$ (i.e., streamflow). It starts by sorting the value of $X_j$ in ascending
order ($j$ indicates the column index of the predictors so that $j \in 1, 2, 3$), and the $Y$ will be reordered
accordingly. Then we go through each instance of $X_j$ from the top to examine each candidate split
point. It should be noted that if there are $K$ instances ($K$=20 in this case), the total number of split
points for $X_j$ will be $K-1$. Any instance $z$ ($z \in 1, 2, ..., K$) in $X_j$ can split the predictor space into two
subspaces as $X_1(j,z) = \{X_{j,1}, X_{j,2}, ..., X_{j,z}\}$ ; $j \in 1, 2, 3$ and $X_2(j,z) = \{X_{j,z+1}, X_{j,z+2}, ..., X_{j,K}\}$ ;





$j \in 1, 2, 3$. The response space $Y$ will be correspondingly divided into two subspaces as $Y_1(z) =$ $\{Y_1, Y_2, ..., Y_z\}$ and $Y_2(z) = \{Y_{z+1}, Y_{z+2}, ..., Y_K\}$ . We also define $R_1 = X_1 \cup Y_1$ and $R_2 = X_2 \cup Y_2$ as

illustrated in Figure 1. To maximize the predictive accuracy, the objective is to find the split point (i.e., $j$ and $z$) that minimize the square error of instances in $Y_1$ and $Y_2$:

$$\sum_{\substack{i \in 1, 2, ..., z; j \in 1, 2, 3 \\ x_i \in X_1(j, z)}} \left( y_i - \overline{y}_{Y_1} \right)^2 + \sum_{\substack{i \in z+1, z+2, ..., 20; j \in 1, 2, 3 \\ x_i \in X_2(j, z)}} \left( y_i - \overline{y}_{Y_2} \right)^2 \qquad (1)$$

where $i$ indicates a particular instance either for $R_1$ or $R_2$; $x_i$ indicates a particular instance in subspace $X_1(j, z)$ or $X_2(j, z)$, $y_i$ indicates a particular instance $y$ in subspace $Y_1$ or $Y_2$, $\overline{y}_{Y_1}$ and $\overline{y}_{Y_2}$

indicate the mean value of $y_i$ in subspaces $Y_1$ and $Y_2$, respectively. Therefore, the split point of the predictor space (i.e., $j$ and $z$) can be obtained by minimizing the equation (1). To remove ties in the argmax, the best split value for $X_j$ is obtained as the average of $X_{j,z}$ and $X_{j,z+1}$ (Scornet, 2020).

*Insert Figure 1 here*

After each split, each of the newly generated subspaces can be further splitted using the same process as long as the number of instances in a subspace is greater than a threshold. This process will be repeated until reaching a stopping criterion, such as a threshold value by which the square errors must be reduced after each split.

The importance score of a particular predictor is measured based on how effective this predictor can reduce the square error in Eq. (1) for the entire tree deduction process (i.e., MDI). In the case of regression, "impurity" reflects the square error of the sample in a subspace (e.g., the larger the square error, the more "impure" the subspace is). The decrease in impurity (DI) for splitting a particular space $s$ is calculated as:

$$DI(j, z, s) = \sum_{i \in 1, 2, ..., k} \left( y_i - \overline{y}_Y \right)^2 - \frac{z}{k} \cdot \sum_{\substack{i \in 1, 2, ..., z \\ x_i \in X_1(j, z)}} \left( y_i - \overline{y}_{Y_1} \right)^2 - \frac{k-z}{k} \cdot \sum_{\substack{i \in z+1, z+2, ..., k \\ x_i \in X_2(j, z)}} \left( y_i - \overline{y}_{Y_2} \right)^2 \quad (2)$$

where $j$ and $z$ are the coordinates for the optimum splitting point of space $s$, $k$ is the number of instances in space $s$ and $\overline{y}_Y$ is the mean value of $y_i$ in space $s$. Therefore, the Mean Decrease in Impurity (MDI) for the variable $X_j$ computed via a decision tree is defined as:



$$MDA\left(X_j\right) = \sum_{\substack{s \in T \\ j=j}} P_s \cdot DI(j,z,s) \tag{3}$$

where $T$ is the total spaces in a tree, $P_s$ is the fraction of instances falling into $s$. In other words, the

MDI of $X_j$ computes the weighted DI related to the splits using the $j_{th}$ predictor.

In Eq. (2), DI is reduced as long as the tree level goes down (i.e., from the top to the bottom level

of the decision tree (shown in Figure 1)). Such treatment naturally assumes that the predictors

considered (for splitting spaces) in lower levels of the tree are less significant than those in upper

levels. This effect is even aggravated by the existence of predictor dependence as which will also

depress the importance scores of independent predictors and increase the positively dependent

ones (Scornet, 2020). As a consequence, predictors considered in lower levels of the tree will only

receive small importance scores and may be neglected by decision-makers. Therefore, the

importance scores obtained from CART is biased towards the predictors considered in early-cut

spaces for obtaining a highest square error reduction. This will mislead the hydrological inference

since some predictors that are considered as less significant in CART (with a limited contribution

in reducing the square error) could be vital for explaining some concerned hydrological behaviors

such as streamflow peaks.

**3 Methodology**

**3.1. Wilks Feature Importance**

For an unbiased estimation of the importance scores for decision trees, the WFI is developed and

illustrated using the same dataset as CART in Figure 1. The fundamental difference between WFI

and MDI comes from the split criterion and procedures used for the tree deduction process. We

will talk about the split criterion first: Recalling the procedure of CART, any possible splits of $X_j$

are examined to find the optimum split point that can minimize the square errors of $Y_1$ and $Y_2$ as

shown in Eq (1). In WFI, the function for finding the optimum split point is achieved by comparing

the two subspaces' likelihood ratio, which is measured through the Wilks' $\Lambda$ statistics. The

optimum value of $\Lambda$ can be used to measure how effective the $X_j$ can differentiate $Y_1$ and $Y_2$.


The calculation process of WFI employs the tree deduction processes of stepwise cluster analysis

(SCA) (Huang, 1992). In SCA, the Wilks' $\Lambda$ statistics (Wilks, 1967;Nath and Pavur, 1985) is used





as the criterion for node splitting, and it is defined as $\Lambda = \dfrac{Det(W)}{Det(B+W)}$ , where $Det(W)$ is the

determinant of a matrix, $W$ and $B$ are the within- and between-group sums of squares and cross

product matrices in a standard one-way analysis of variance, respectively. To define $W$ and $B$, let

$Y_1$ and $Y_2$ contain $z$ and ($k$-$z$) instances, respectively. We have the following vectors:

$y_{Y_1}(i) = \left\{y_{1,i}, y_{2,i}, ..., y_{d,i}\right\}$, $i = 1, 2, ..., z$, and $y_{Y_2}(i) = \left\{y_{1,i}, y_{2,i}, ..., y_{d,i}\right\}$, $i = z+1, z+2, ..., k$, where $d$

is the number of columns of $Y_1$ and $Y_2$ ($d$=1 in this case). Then the $W$ and $B$ can be given by:

$$B = \sum_{i=1}^{z}\left[y_{Y_1}(i) - \overline{y_{Y_1}}\right]' \cdot \left[y_{Y_1}(i) - \overline{y_{Y_1}}\right] + \sum_{i=1}^{k-z}\left[y_{Y_2}(i) - \overline{y_{Y_2}}\right]' \cdot \left[y_{Y_2}(i) - \overline{y_{Y_2}}\right] \tag{4}$$

$$W = \frac{z \cdot (k-z)}{k}(\overline{y_{Y_1}} - \overline{y_{Y_2}})' \cdot (\overline{y_{Y_1}} - \overline{y_{Y_2}}) \tag{5}$$

The test statistics $\Lambda$ represent the likelihood ratio of two subspaces, the smaller $\Lambda$ value

representing a larger difference between the sample means of $Y_1$ and $Y_2$. The distribution of $\Lambda$ is

approximated by Rao's $F$-approximation ($R$-statistic), which is defined as:

$$R = \frac{1 - \Lambda^{1/S}}{\Lambda^{1/S}} \cdot \frac{Z \cdot S - d \cdot (m-1)/2 + 1}{d \cdot (m-1)} \tag{6}$$

$$Z = k - 1 - (d + m)/2 \tag{7}$$

$$S = \frac{d^2 \cdot (m-1)^2 - 4}{d^2 + (m-1)^2 - 5} \tag{8}$$

where statistic $R$ is distributed approximately as an $F$-variate with $n_1 = d \cdot (m-1)$ and

$n_2 = d \cdot (m-1)/2 + 1$ degrees of freedom; $m$ is the number of groups. Since the number of groups

is two in this study, an exact $F$-test is possibly performed based on the following Wilks' $\Lambda$ criterion

be:

$$F(d, k-d-1) = \frac{1 - \Lambda}{\Lambda} \cdot \frac{k-d-1}{d} \tag{9}$$

Therefore, the sample means of the two subspaces can be compared for examining significant

differences through the $F$-test. The null hypothesis would be $H_0$: $\mu(Y_1) = \mu(Y_2)$ versus the

alternative hypothesis $H_1$: $\mu(Y_1) \neq \mu(Y_2)$, where $\mu(Y_1)$ and $\mu(Y_2)$ are population means of $Y_1$ and

$Y_2$, respectively. Let the significance level be $\alpha$ (which is set as 0.05 in this study), the split criterion





would be: $F_{cal} \geq F_{\alpha}$ and $H_0$ are false, which implies that the difference between two subspaces is significant thus they should be splitted.

By far, the splitting criteria of CART and SCA is compared, the second difference of these two algorithom is the tree deduction procedure. In CART, the splitting process will be repeated until any of the newly generated subspace can no longer be splitted. While in SCA, once all the nodes in the current stage have been examined for the splitting process, merging process will be followed in next stage as illustrated in Figure 1. The merging process will compare any pairs of nodes based
on Wilks' $\Lambda$ value to test if they can be merged (i.e., for $F_{cal} < F_{\alpha}$ and $H_0$ are true, which indicates that these two subspaces have no significant difference thus should be merged). Such splitting and merging processes are iteratively performed until no node can be further split or merged. Once an SCA tree is built, the WFI for the variable $X_j$ computed via an SCA tree is defined as:

$$WFI\left(X_j\right) = \sum_{\substack{s \in T \\ j = j}} P_s \cdot \left(1 - \Lambda(j, z, s)\right) \tag{10}$$

where $\Lambda(j, z, s)$ denotes the value of $\Lambda$ obtained at the optimum splitting point of space $s$ with column and row coordinates $j$ and $z$, respectively. Similar to the calculation of MDI in Eq. (3), the WFI for $X_j$ computes the weighted $(1-\Lambda)$ value related to the splits using the $j_{th}$ predictor.

The major advantage of WFI over MDI is that every spliting and merging action by WFI is
evaluated based on Wilk's test-statistics with the significance level $\alpha$ set equals 0.05, which greatly reduced probabilities that the two child-nodes are splitted due to chance. However, the node splitting actions in MDI approach is purely based on square error, which can potentially lead to overfitting. Moreover, the WFI can provide an unbiased estimation of important scores compared with MDI, since the values of $(1-\Lambda)$ do not necessarily decline as long as the tree level goes down
(as shown in Figure 1). Therefore, the $X_j$ that is mostly considered in latter splits is still possible to have a higher importance score than it in early splits as long as the $\Lambda$ values for those splits are small enough. As the consequence, predictors that contribute to specific model predictions can be identified by WFI but may be overlooked by MDI.





### 3.2 Stepwise Clustered Ensemble and Posterior-Informed WFI

Similar to MDI, the WFI performs better under the regression tree ensemble (RTE) framework since the randomized predictors ensure enough tree diversity, which in turn, leads to more balanced importance scores (Scornet, 2020). The concept of RTE is to grow trees following a random subset of input predictors sampled without replacement and a slightly different set of instances drawn randomly from the training data with replacement. As the ensemble members (trees) increase, the

non-linear relationships between predictors and responses become increasingly stable and the prediction can thus be more robust and accurate (Breiman, 2001;Zhang et al., 2018). Therefore, in this study, the WFI will be evaluated under the RTE framework. The SCA ensemble (SCE) will contain $N$ (which is the number of trees) different sets of important scores, the ensemble (i.e., average) of these sets of important scores is assumed to be more robust than the individual one.


By far, the importance scores from WFI can provide a global perspective of how significant a particular predictor is related to model predictions. A more intriguing question could be how does the significance of a predictor vary in response to the variations of streamflow? To this end, the proposed WFI is extended under the SCE framework through the Bayesian model averaging (BMA)

approach. The BMA (Raftery et al., 2005) aims to provide the likelihoods (i.e., BMA weights) of each ensemble member that can best match the observations. In this study, each SCA tree is considered as an ensemble member of BMA. The BMA algorithm is then applied to a spectrum of streamflow quantile ranges, which will lead to sets of BMA weights. Each set of BMA weights reflects the likelihoods of each SCA tree being the best prediction over a particular streamflow

quantile ranges. By combining the sets of BMA weights and the sets of importance scores, the posterior-informed Wilks feature importance (PWFI) will be able to emphaze the importance scores at a particular quantile range. The procedure of PWFI is given as follows:

Consider a set of SCA trees $T$, where $T = \{T_1, T_2, ..., T_n\}$, and $n$ is the total number of trees; $f$ is the

number of predictors used to build each SCA tree; $F$ is the total number of predictors used in the model, where $f \in F$; $M$ is the number of interested streamflow quantile ranges, and $m$ is a particular quantile range ($m \in 1, ..., M$); $I(i, j)$ is the importance score for the $i_{th}$ SCA tree and its $j_{th}$ predictor; $E(\cdot)$ denotes the function for an SCA tree to make a prediction.





(1)    Calculate sets of BMA weights of $E(T_i)$ under the quantile range $m$, denoted as $B(i, m)$,
where $i \in 1, …, n$ and $m \in 1, …, M$.

(2)    Calculate sets of Wilks importance scores $WI(i, j, m)$ for the $i_{th}$ SCA tree under the quantile
range $m$, to obtain $WI(i, j, m) = B(i, m) \times I(i, j)$, where $i \in 1, …, n, j \in 1, …, f$ and $m \in 1, …, M$.

(3)    Aggregate (i.e., average) the $WI(i, j, m)$ along the coordinate $i$ (across all the SCA trees),

to obtain $AWI(j, m) = \sum_{i=1}^{n} WI(i, j, m) \Big/ \sum_{i=1}^{n} Count(T_i, j)$, where $j \in 1, …, F, m \in 1, …, M$ and $Count(\cdot)$

is a function for testing whether the $i_{th}$ SCA tree uses the $j_{th}$ predictor (1 means **yes** and 0 means
**no**).

(4)    Normalize the $AWI(j, m)$ to the [0,1] range along the coordinate $m$, so that the PWFI($j, m$)

$$= AWI(j, m) \Big/ \sum_{m=1}^{M} AWI(j, m); j \in 1, …, F; m \in 1, …, M .$$

Therefore, the above procedures "downscale" the Wilks importance scores into streamflow
quantile ranges of interest, which facilitate the investigation of the effects from varied predictor
characteristics. The detailed training process of BMA follows the procedure of Duan et al. (2007).

## 4. Application of WFI

### 4.1. Study Area and Data

Three irrigated watersheds located in the alluvial plain of the Yellow River in China were selected
to test the capability of the proposed WFI method (Figure 2). These watersheds share a total area
of 4,905 km$^2$, consisting of 52% irrigated land, 17% residential area, 15% desert, 12% forested
land, and 4% water surface. The landscape of the study area is characterized by an extremely flat
surface with an average slope ranging from 1:4000 to 1:8000, with mostly highly permeable soil
(sandy loam). The climatic condition of the study area is characterized by extreme arid
environments with annual precipitation ranging from 180 to 200 mm, and annual potential
evaporation ranging from 1,100 to 1,600 mm (Yang et al., 2015).

*Insert Figure 2 here*

Initial catchment conditions were considered to improve the model performance. Specifically,
moving sums of daily precipitation, temperature and evaporation timeseries over multiple time
periods $\delta_{P,T,E} = [1, 3, 5]$ prior to the date of predictions were set as predictors to reflect the





antecedent watershed conditions. Similary, the moving window for daily irrigation timeseries $\delta_I =$ [1, 3, 5, 7, 15, 30]. In addition, daily groundwater level data are used as additional predictors to reflect the baseflow conditions of the catchments. The daily timeseries data were divided into two

subsets, including one from 2001/01/01 to 2011/12/31 and the other from 2012/01/01 to 2015/12/31 for model training and prediction, respectively. Table 1 list the weather, rain and groundwater stations used for each basin. Owing to the different irrigation schedules in spring and winter, the streamflow processes show distinct behaviors in terms of flow magnitude and duration. To analyze such temporal variations, the hydrological processes for Spring-Summer (April to

September) and Autumn-Winter (October to March) were examined separately.

***Insert Table 1 here***

Five hyperparameters need to be determined to train the SCE, including the significance level ($\alpha$) used for the *F*-test during the node splitting process, the number of trees (*Ntree*), the minimum

number of samples in a node (*Nmin*) for a splitting action, and the number/ratio of predictors in a subspace (*Mtry*). In this study, we set the $\alpha$ value for 0.05 as suggested by Huang (1992). The *Ntree* was set as 200, after which no further improvement in model validation accuracy can be achieved. The *Nmin* was set as 5 to ensure the rare events can be identified. The *Mtry* was set as 50% as suggested by Barandiaran (1998), indicating half of the predictors are selected in each SCA

tree. Similar to the RF, cross-validation is not required for the SCE since one third of the training data will not be used for training each SCA tree and these out-of-bag (OOB) data will be used as the validation dataset. We use the R package "randomForest" (Liaw and Wiener, 2002) for training the RF model with the default settings. The XGB training is based on 10-fold cross-validation scheme using the R package "XGBoost" (Chen et al., 2015).

**4.2. WFI Evaluation**

The performance of WFI will be evaluated and compared against the permutation feature importance (PFI) method (applied to RF and SCE model) and the mean decrease in impurity (MDI) method (applied to RF and XGB model). The PFI method follows the procedure of Molnar (2020): Assume a trained model $\boldsymbol{M}$ with $\boldsymbol{p}$ predictors, predictor matrix $\boldsymbol{X}$, response vector $\boldsymbol{Y}$, predicted

vector $\boldsymbol{Y'}$, and an error measure $\boldsymbol{L(Y, Y')}$; (1) calculate the original model error based on the validation dataset (out-of-bag dataset in our case) $\boldsymbol{e_{org}(M) = L(Y, M(X))}$; (2) for each predictor





$j \in 1,...,p$, (i) generate permuted predictor matrix $X_{perm,j}$ by duplicating $X$ and shuffling the values of predictor $X_j$, (ii) estimate error $e_{perm,j} = L(Y, M(X_{perm,j}))$, and (iii) calculate PFI of predictor $j$ as $PFI_j = e_{perm,j}/e_{org}(M)$. The MDI follows the procedure of the conventional random forest approach

as illustrated in section 2. All three interpretation methods will be evaluated through recursive feature elimination (RFE) (Guyon et al., 2002) as follows: (1) train three models (i.e., SCE, RF and XGB) with all predictors; (2) calculate the importance scores using the three interpretation methods (i.e., PFI, MDI and WFI) embedded in their corresponding models (i.e., the PFI method will be applied to RF and SCE; MDI will be applied to RF and XGB; WFI is only available to

SCE); (3) exclude the three least relevant predictors for each set of the important scores obtained in step 2; (4) retrain the models using the remaining predictors in step 3; (5) repeat step 2 to 4 until the number of predictors reaching a minimum threshold (which was set as five in this study).

The evaluation of the three interpretation methods is based on the two aspects as interpretation

accuracy and robustness. The interpretation accuracy (for a feature importance method) is defined as the difference between the predictive accuracy achieved by the most relevant predictors (identified by this feature importance method) and that achieved by all the considered predictors. In detail, the evaluation of interpretation accuracy starts from setting the evaluation metrics. In this study, we use the RMSE and adjusted $R^2$ as the evaluation metrics for the model prediction. The

adjusted $R^2$ has been used instead of $R^2$ because the previous metric can consider the number of predictors. The adjusted $R^2$ is defined as:

$$adj\ R^2 = 1 - \frac{\left(1 - R^2\right)\left(N - 1\right)}{N - P - 1} \tag{11}$$

where $P$ is the number of predictors and $N$ is the number of instances. By evaluating the RMSE and adjusted $R^2$ after each RFE iteration, we can observe the iterative reduction of the model

predictive accuracy using the remaining predictors. For the remaining predictors that are more relevant to the model predictions, a smaller increment (or reduction) in RMSE (or adjusted $R^2$) will be observed, thus the higher interpretation accuracy will be achieved.



The interpretation robustness means the relative changes of importance scores for the most relevant predictors in response to the reduction of irrelevant predictors. The interpretation robustness is evaluated as follows: for each RFE iteration, the number of each set of predictors will be reduced by three. The reduced sets of importance scores will be compared with those in previous iterations in terms of the relative changes in score values. Given the total shares of the importance scores for any set equals 100%, the RFE will increase the share of importance scores for the most relevant predictors after each iteration. Therefore, a monotonically increasing trend for the importance scores of a particular predictor should be expected in response to the iterative reduction of irrelevant predictors. In this study, the monotonicity is examined by using the Spearman's rank correlation coefficient (i.e., Spearman's ρ), which is commonly used to test the statistical dependence between the rankings of two variables, and is defined as:

$$\rho = \frac{\sum_i \left( RX_i - \overline{RX} \right)\left( RY_i - \overline{RY} \right)}{\sqrt{\sum_i \left( RX_i - \overline{RX} \right)^2 \left( RY_i - \overline{RY} \right)^2}} \tag{12}$$

where $\boldsymbol{RX_i}$ is the ranks of variables $\boldsymbol{X}$ for the $\boldsymbol{i}_{\text{th}}$ RFE iteration and $\boldsymbol{RY_i}$ is the number of selected predictors for the $\boldsymbol{i}_{\text{th}}$ RFE iteration; $\overline{RX}$ and $\overline{RY}$ are the means of $\boldsymbol{RX_i}$ and $\boldsymbol{RY_i}$, respectively. A larger Spearman's $\boldsymbol{\rho}$ indicates the importance score for a predictor will increase along with the reduction of irrelevant predictors, which therefore will lead to relatively more robust importance scores in terms of the pattern of importance scores.

## 5. Results

### 5.1. Interpretation Accuracy and Robustness

Generally, all three algorithms delivered reasonable predictive accuracy (by using all considered predictors) across all irrigated watersheds and seasons (Table 2). The SCE approached the best overall predictive accuracy for the testing dataset. When compared with RF, the SCE has a smaller drop in accuracy from training/validation to testing, which indicates the SCE algorithm can better address overfitting.

***Insert Table 2 here***





400 Compared with other benchmark algorithms and interpretation methods, WFI shows the best interpretation accuracy as it has the overall smallest average reduction in predictive accuracy (considering all the RFE iterations) for the testing dataset (Figure 3). The iterative reductions in accuracy for training, validation and test datasets are listed in Figure S1, S2 and Figure 4, respectively. Surprisingly, we found that the predictors selected by SCE-WFI lead to even higher

405 predictive accuracy on the testing dataset than the SCE-PFI selected ones. The possible reason is that the PFI method can only consider the effect of one predictor at a time, thus the interactions (i.e., quadratic terms) between the considered predictor and the rest predictors are overlooked. The WFI method on the other hand naturally considers all the interactions among predictors in the tree deduction process, thus the importance scores are considered to be more comprehensive than those

410 generated by the PFI method. Similar evidence can also be found between RF-Purity (i.e., MDI method) and RF-PFI methods: the average reduction in accuracy for RF-Purity is less than that for RF-PFI under the testing dataset.

***Insert Figure 3 here***
***Insert Figure 4 here***


Comparative studies among the three interpretation methods illustrate overfitting can greatly affect predictive and interpretation accuracy. For instance, RF-PFI owns the lowest average reduction in adjusted $R^2$ among all models on the training dataset, while its value becomes the highest on the testing dataset. A smaller reduction in accuracy means the retained predictors are more informative

420 in describing the complex relationships of hydrological processes. Based on this, SCE-WFI can provide the most informative predictors among all considered models because it suffers the least from overfitting. The results also indicate that the XGB algorithm suffers less from overfitting compared with RF. Nevertheless, the predictive accuracy for the XGB algorithm is not as good as it for RF or SCE (Table 2).

425      ***Insert Table 2 here***

The Spearman's ρ values for the most relevant predictors (i.e., selected by the last iteration of RFE) across all drainage basins and seasons illustrate the robustness of all three interpretation methods embedded in different models. The results indicate the relative importance of a particular predictor

430 increase in response to the reduction of irrelevant predictors (Figure 5 and Figure S3-S6).





Compared with other interpretation methods and ML algorithms, the SCE-WFI owns the highest absolute Spearman's ρ values for the majority of the cases (Figure 6). This indicates the reduction of irrelevant predictors would greatly influence the importance scores obtained by PFI and MDI. This challenges the application of the PFI and MDI since the removal of irrelevant predictors

cannot guarantee the same or similar level of hydrological inference (i.e., the relative importance scores may vary distinctly according to the reduction of irrelevant predictors). In contrast, the WFI method provides more stable relative importance scores and will lead to more consistent hydrological inferences.

***Insert Figure 5 here***

***Insert Figure 6 here***

### 5.2. Insights Toward Hydrological Processes

To explore the relationships between the hydrological responses and their driving forces, the importance scores were aggregated and analyzed according to different types (i.e., precipitation, irrigation, evaporation, etc.). We chose the models with the smallest RMSE (among all the RFE

iterations) on the testing dataset to investigate the relationships between importance scores and hydrological processes. The results indicate the importance scores differed significantly according to the algorithms and interpretation methods used (Figure 7). In particular, the aggregated predictor P1 (i.e., precipitation of the current timestep from all spatial locations) owns positive contributions (in reducing the RMSE) for SCE-WFI in the Spring irrigations, while it has no contribution for

other ML algorithms and interpretation methods. To investigate whether the predictors identified by WFI are also meaningful to other algorithms, we reinserted the predictors assoicated with P1 into the best-performance RF and XGB models. Surprisingly, we found both models showing slightly improved predictive accuracy (i.e., RMSE and adjusted $R^2$) for Spring irrigations across all drainage basins on the testing dataset (Table 3). This finding reveals that even though the P1

has no contribution in improving the predictive accuracy on the training dataset, it can still distinguish different hydrological behavior (i.e., with a small Wilk's Λ value) and have the potential to improve the model performance on the testing dataset. In fact, the time of concentration for these basins are usually less than one day if the storm falls near the outlets of the irrigation basins. This fact proves the above hydrological inference is reasonable.

***Insert Figure 7 here***



*Insert Table 3 here*

It should be noted that the variation of importance scores among predictors for the WFI method is
much smaller than it for other feature importance methods. This is caused by the nature of Wilk's
Λ: In the node splitting process, a predictor that significantly increases the predictive accuracy
may not necessarily have a strong separative power (i.e., a small Λ value) to differentiate two
potential sub-spaces. As a consequence, such a predictor could gain a relatively higher importance
score for accuracy-based interpretation methods than the WFI. However, predictors identified by
accuracy-based interpretation methods maybe subject to overfitting, which does not guarantee a
valid inference on the testing dataset. The WFI method (which evaluates every spliting and
merging action based on Wilk's test-statistics with the predefined significance level $\alpha$) is less likely
to be overfitted and expected to generate more reliable importance scores.

Figure 8 and S8 depict the varying roles of a predictor played in the hydrological processes, that
is, streamflows at 25%, 50%, 75%, 90%, 95%, 99% and 100% percentiles. We found the P3 and
P5 (i.e., 3 and 5-day accumulative precipitation) have higher importance scores for peak flows
than those for low flows during the spring irrigation periods, while no significant trends can be
observed for P1. This is probably because the accumulative precipitation bears the information of
both antecedent watershed conditions and the storms, which are the keys to the formation of
streamflow peaks. Significant trends on importance scores also can be found for some irrigation-
related factors. For instance, the I1 and I3 (i.e., one- and three-day accumulative irrigation) in the
first drainage area share relatively higher importance scores on low flows than those for the high
and peak flows. This probably due to the majority of the irrigated lands in the first drainage area
being paddy fields, which require flood irrigation to soak the rice fields (which are usually bunded)
for a few days. Once the unsaturated zone of the soil becomes saturated (which usually happens
two weeks after the beginning of irrigation), the groundwater table will be elevated and the
irrigation factors such as I15 and I30 will increase their dominance on peak flows.

*Insert Figure 8 here*

Owing to the different characteristics of the study watersheds, the same factor may behave
distinctively according to the landuse, irrigation schedule and cropping pattern. The third drainage



basin, for example, shows a decreasing (or increasing) trend of importance scores for I30 (or P5) as the flow magnitude increases. This indicates that the peak flows in this area are probably caused by excessive rainfall rather than long-term irrigation. This is quite different from the first irrigated

area. In fact, most of the irrigated lands in the third drainage area grow corn which does not require to be soaked as rice does. Therefore, the irrigated water in this area will drain faster than that in the first irrigated area. Moreover, the third drainage baisn includes more mountainous area than other basins, which allows a shorter time of concentration, making the precipitation the dominant factor in the peak flows.

**6. Discussion**

Previous studies indicated that equifinality is a major challenge for hydrological inference using conventional machine learning approaches such as RF and MLP (Schmidt et al., 2020). Studies from Schmidt et al. (2020) mentioned that the patterns of importance scores (achieved by PFI) may vary significantly according to different algorithms. Their conclusions were verified by our

results and at the same time, we also found that equifinality exists within an algorithm: the reduction of irrelevant predictors (from the full model) may lead to the same or similar predictive accuracy (Figure 4) but the importance score patterns across iterations could be distinctive (Figure 7). To differentiate these two types of equifinality, we name the equifinality between algorithms as type-A equifinality and name the one within an algorithm as type-B equifinality. The

comparative analysis in our study indicates the type-B equifinality can be addressed by the WFI method, which produces more robust importance scores than its counterparts. However, the WFI is a model-specific method, which means it cannot be extended to other algorithms and is thus unable to deal with the type-A equifinality. Nevertheless, the proposed WFI embedded in the SCE model provides a new and reliable solution for hydrological inference.


The comparative studies between SCE-WFI and SCE-PFI indicate that the high robustness of importance scores by SCE-WFI may come either from the decision trees or the WFI method. However, when importance scores are compared between RF-Purity with RF-PFI, the RF-Purity importance scores are no more robust than the RF-PFI one's. This leads to a deduction that the

high robustness of importance scores by SCE-WFI is due to the WFI method rather than decision trees. The reason MDI does not share the merits of WFI is probabally because the tree deduction


processes for RF and XGB are both associated with predictive accuracy (i.e., square errors), which can lead to overfitting. The most relevant predictors determined by such "overfitted" trees are biased towards predictors for the highest square error reduction in training dataset, and may not
guarantee a reasonable inference on the testing dataset. The WFI approach, on the other hand, evaluates every spliting and merging action based on Wilk's test-statistics rather than accounting for predictive accuracy, thus generates relatively unbiased importance scores. A sound hydrological inference should derived from a "general solution" rather than a "special solution". The WFI approach obtained such a "general solution" for hydrological inference by learning the
differences among complex hydrological behaviors. However, the MDI and PFI methods are more likely to obtain a "special solution" for hydrological inference since they are based on predictive accuracy (which is always case sensitive and may lead to overfitting). This challenges the MDI and PFI to reflect the whole picture of hydrological process.

***Insert Figure 9 here***


The analysis of BMA weights for the posterior-informed WFI indicates that the BMA algorithm can select the most promising SCE decision trees for each flow quantile range (Figure 9 and S9 to S13). The evenly distributed BMA weights under each flow quantile range illustrates the BMA algorithm could obtain the global optimum solution for its objective (i.e., smallest RMSE). The
varying BMA weights among decision trees highlights the diversities of random predictors selected for building these decision trees.

### 7. Conclusions

The Wilk's feature importance was developed to improve the interpretability of decision trees and regression tree ensembles. Our results indicate the proposed WFI provides a more robust
hydrological inference, compared with the well-known PFI method and MDI method. In addition, we found WFI can identify more informative predictors, compared with PFI and MDI in terms of predictive accuracy (i.e., adjusted $R^2$ and RMSE). With the provisions of the BMA algorithm, the posterior information allowed the WFI methods to downscale the global importance scores to local ones. The localized importance scores can reflect the varying characteristics of a predictor
involved in the hydrological processes.





There are three main achievements of the proposed WFI in hydrology: firstly, the issue of equifinality that exists in conventional statistical models can be partially addressed by using the proposed WFI method; secondly, some critical predictors that may be overlooked by conventional feature importance methods can be captured through the WFI; thirdly, the posterior-informed WFI can help to gain insights into some hydrological behaviours.

Although a complete description of all the decision trees within the model is infeasible, the proposed WFI could be a step closer for hydrologists to get a preliminary understanding of the hydrological process through machine learning. However, several challenges still exist in the current interpretation approach such as how to find the best balance amongst the model complexity, performance, and interpretability. A complex model may yield higher performance skills than a simple model, but at the same time, will introduce the multicollinearity problem, which in turn will hamper the model interpretability. Moreover, current applications of importance scores are still limited. As interpretable machine learning models continue to mature, the potential benefits of hydrological inference could be promising if the importance scores can be associated with physically-based hydrological models.

**Code and data availability.** The climatic data are available on the data repository of China meteorological data service center (http://data.cma.cn/en). The hydrological data and code for the numeric case can be accessed from Zenodo repository (https://doi.org/10.5281/zenodo.4387068). The entire model code for this study can be obtained upon email request to the corresponding author.

**Author contribution.** Kailong Li designed the research under the supervision of Guohe Huang. Kailong Li carried out the research, developed the model code and performed the simulations. Kailong Li prepared the manuscript with contributions from Guohe Huang and Brian Baetz.

**Competing interests:** The authors declare that they have no conflict of interest.

**Acknolwdgement.** We appreciate Ningxia Water Conservancy for offering the streamflow, groundwater and irrigation data and related help. We are also very grateful for the helpful inputs from the Editor and anonymous reviewers



**Financial support.** This research was supported by Canada Research Chair Program, Natural Science and Engineering Research Council of Canada, Western Economic Diversification (15269), and MITACS.

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

**Figures:**

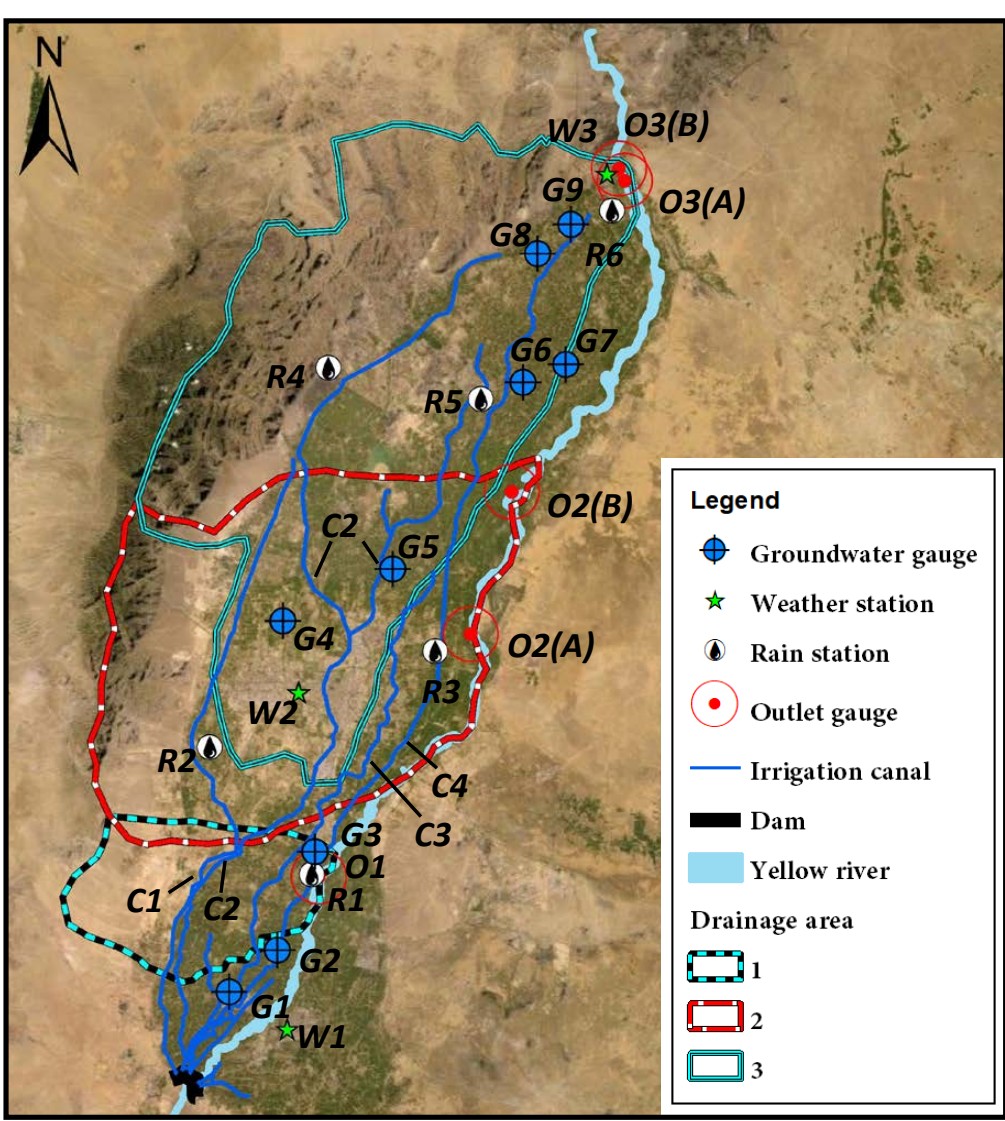


**Figure 1**: Map of the study area. Note: due to the extreme flat surface, three interconnected drainage areas are approximately delineated. In this map, *G* indicates groundwater gauges, *W*





indicates weather stations, *R* indicates rain stations, *C* indicates irrigation canals and *O* indicates drainage outlets. Both 2nd and 3rd drainage areas contain two crisscrossed drainages with strong

hydrological connections. The map was created using ArcGIS software (Esri Inc. 2020).

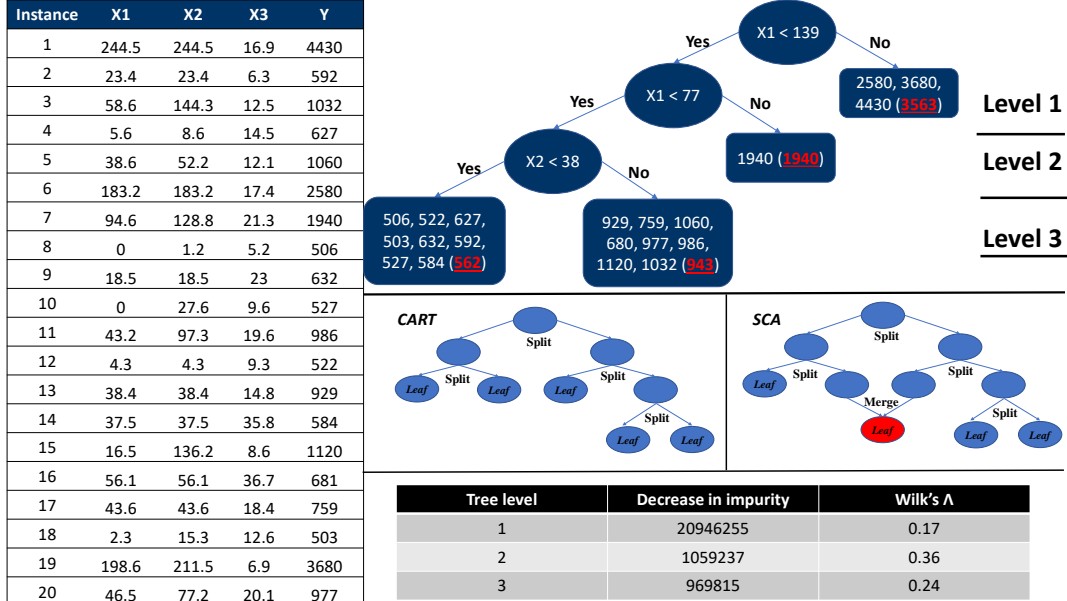

**Figure 2**: Table on the left is a numeric hydrological dataset; figure on the top right is the tree deduction process for both CART and SCA with the dataset (note: the highlighted numbers in

brankets of the leaf-nodes are the mean response values of those nodes; in this special case, the two algorithms share the same node splitting rules, however, for most real-world cases, they lead to different decision trees); figure on the middle right illustrates the typical difference of deduction process between CART and SCA (not related to the case); table on the bottom right is the statistic summaries for CART and SCA of this case.






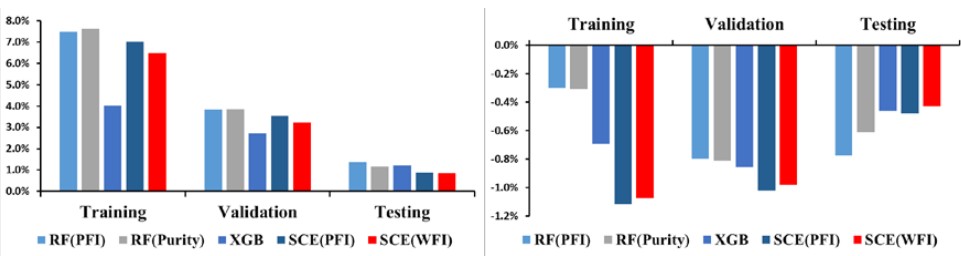

**Figure 3**: Reduction in accuracy (RMSE is on the left and adjusted $R^2$ is on the right) on average across three drainage basins, seasons and RFE iterations.

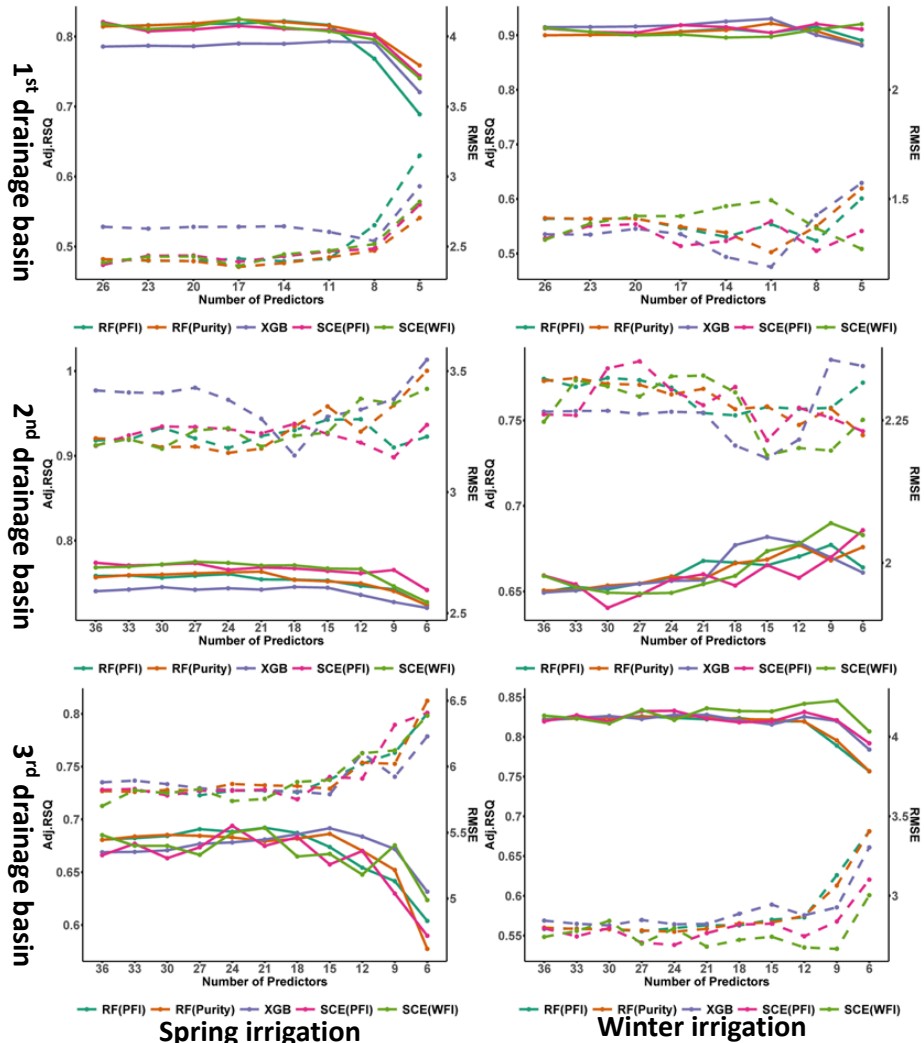





**Figure 4:** Iterative reduction in accuracy for the testing dataset. Note: The solid lines indicate adjusted R$^2$, while the dashed lines represent RMSE.

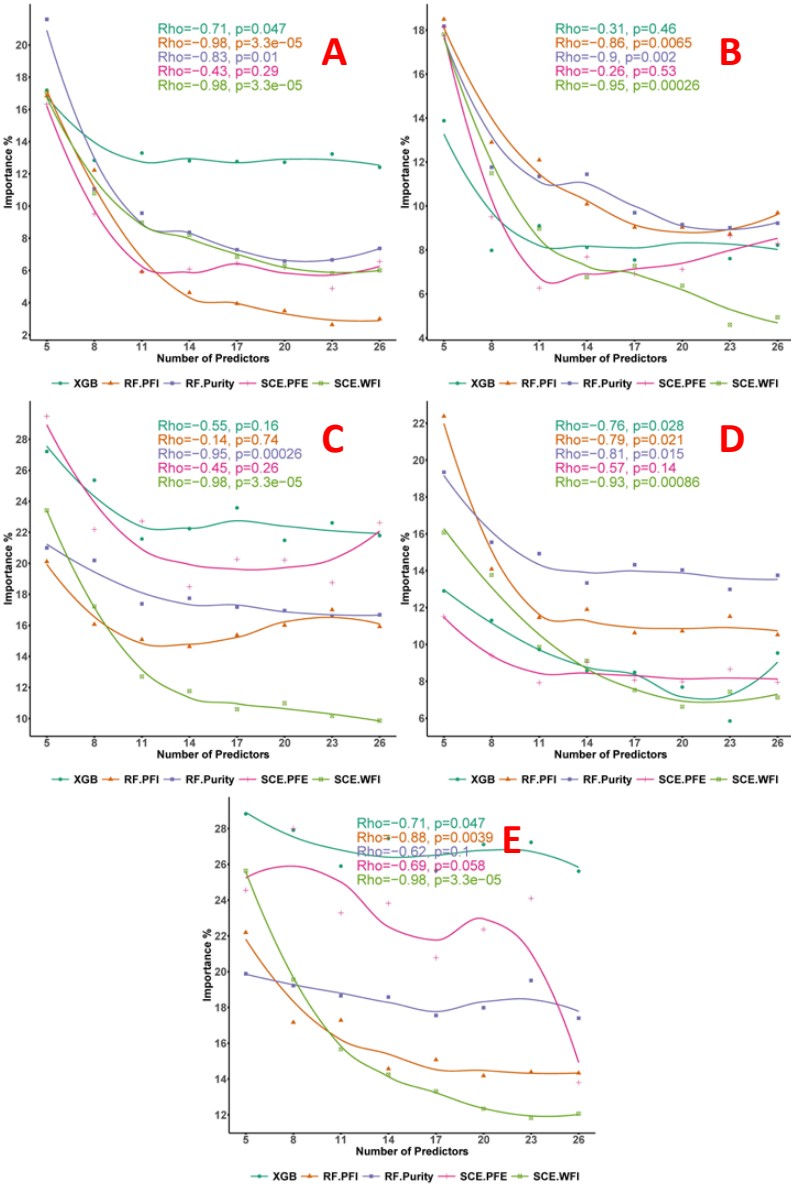

**Figure 5**: Spearman's ρ values for the first drainage basin during the Spring irrigation period. The p-value means how likely it is that the observed correlation is due to chance. Small p-values





indicate strong evidence for the observed correlations. Capital letters from A to E represent the five most relevant features identified by different models.

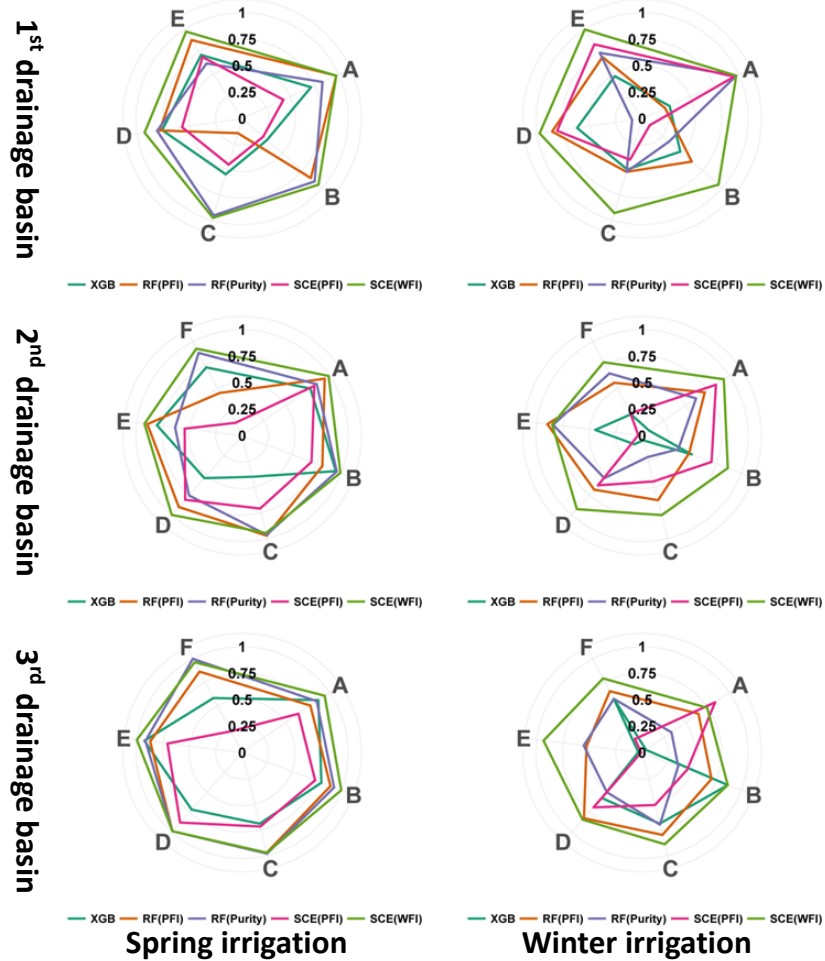

**Figure 6**: Spearman's ρ values for all three basins and irrigation seasons. Note: the RFE process keeps at least five and up to seven of the most relevant predictors in the last iteration, according to

the remainder of total considered predictors divided by three. Capital letters from A to F represent the most relevant features identified by different models.



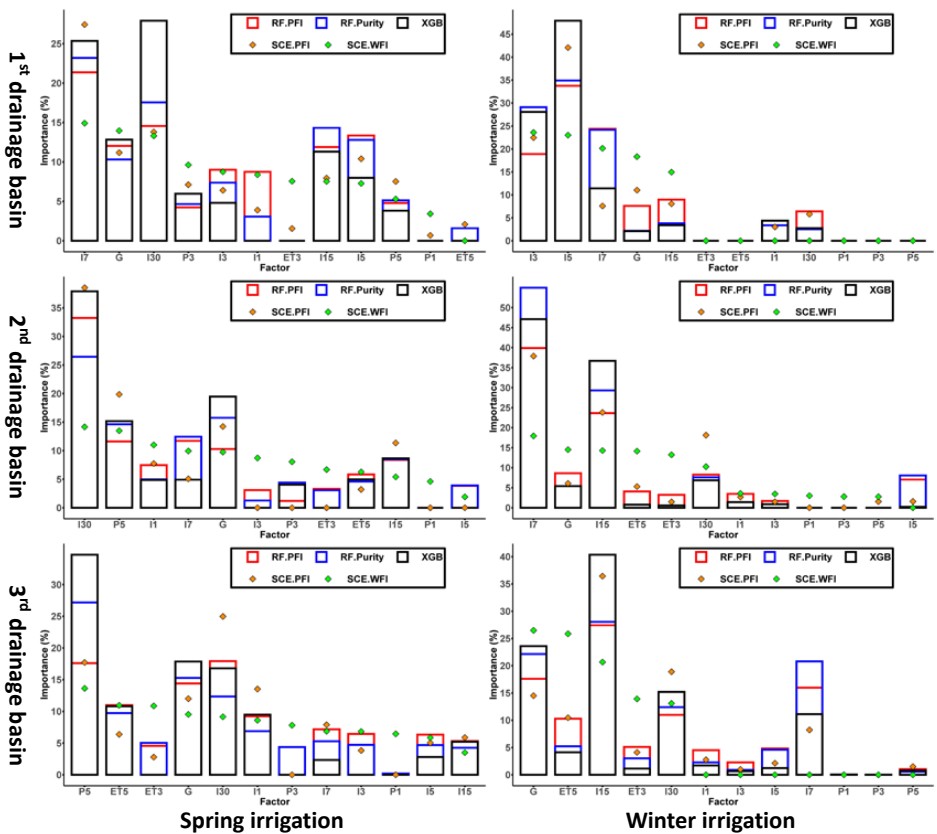

**Figure 7**: Importance scores aggregated by predictor types. Note: each type of predictor includes predictors from all considered spatial locations. For example, P1 includes predictors for all the considered climatic stations with 1-day precipitation. The importance score of P1 is the average of the importance score from the predictors of P1.



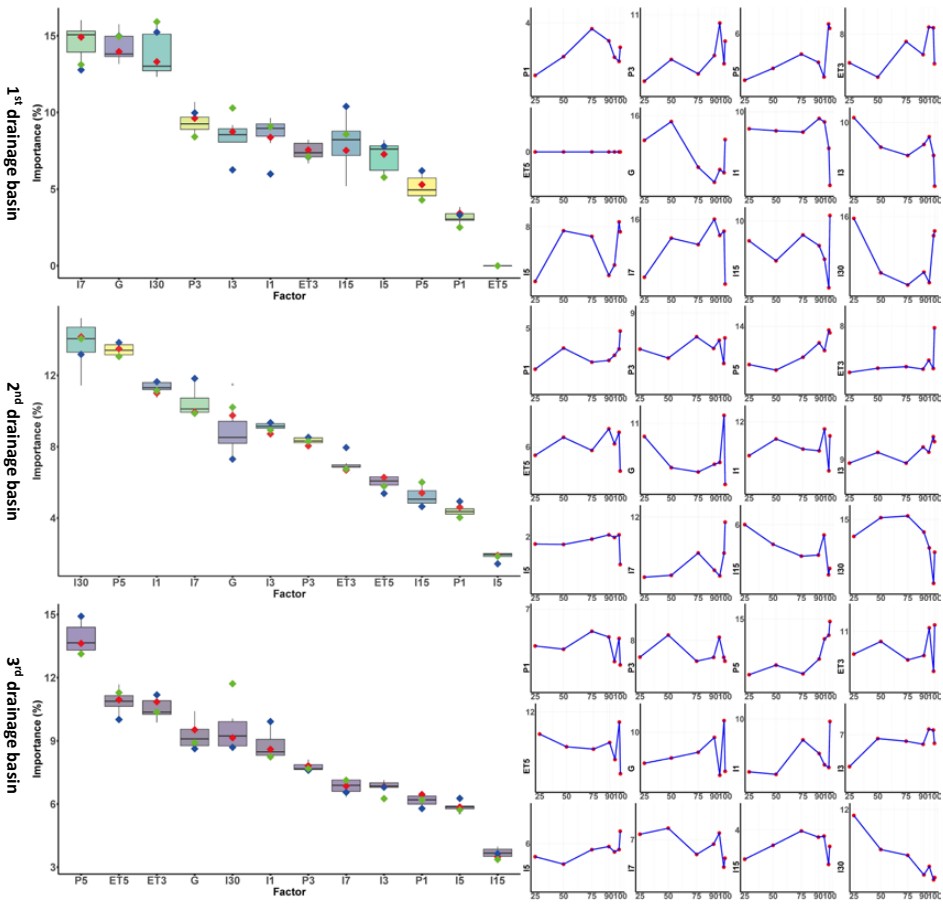

**Figure 8**: Posterior-informed importance scores at different flow quantile intervals for the Spring irrigation period. Note: the importance scores at different quantile intervals are represented as box and whisker plots, the mean feature importance (measured using normal WFI method) is represented as red diamonds. The green and blue diamonds are feature importance at 25[th] and 100[th] flow quantiles, respectively. The line plots on the right side represent how importance scores vary along with the changes in flow quantile. The "x" and "y" axis of the line plots are flow quantiles at 25, 50, 75, 95, 95, 99 and 100 (%), and feature importance (%), respectively.

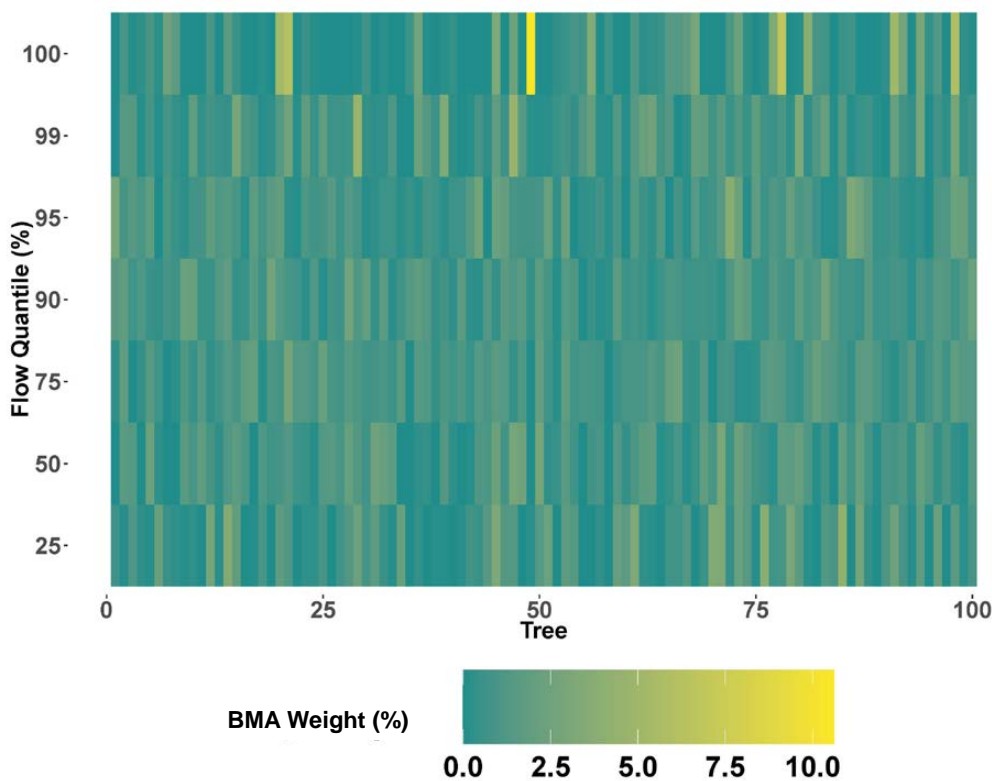


**Figure 9**: BMA weights of SCE decision trees for the first drainage basin at the Spring irrigation period under seven flow quantile ranges.





**Tables:**

**Table 1**: Weather, rain and groundwater gauges, and irrigation cannels used in each irrigation basin.

| Drainage area | Stations/cannels | outlets |
|---|---|---|
| 1st Drainage area | *C1*, *C2*, *C3*, *W1*, *R1*, *G1*, *G2*, *G3* | *O1* |
| 2nd Drainage area | *C1*, *C2*, *C3*, *C4*, *W2*, *R2*, *R3*, *R5*, *G4*, *G5* | *O2(A)+ O2(B)* |
| 3rd Drainage area | *C1*, *C2*, *C4*, *W2*, *W3*, *R4*, *R5*, *R6*, *G4*, *G5*, *G6*, *G7 G8*, *G9* | *O3(A)+ O3(B)* |

Note: Streamflow for each drainage area is predicted as the sum of the gauged streamflows within this area.

**Table 2**: The adjusted $R^2$ for three algorithms with all considered predictors.

| Dataset | Season | SCE | RF | XGB |
|---|---|---|---|---|
| Training | 1st Spring | 0.94 | 0.98 | 0.90 |
| | 1st Winter | 0.98 | 0.99 | 0.98 |
| | 2nd Spring | 0.94 | 0.98 | 0.89 |
| | 2nd Winter | 0.98 | 0.99 | 0.97 |
| | 3rd Spring | 0.94 | 0.98 | 0.87 |
| | 3rd Winter | 0.98 | 0.99 | 0.97 |
| Validation | 1st Spring | 0.87 | 0.88 | 0.83 |
| | 1st Winter | 0.94 | 0.95 | 0.94 |
| | 2nd Spring | 0.86 | 0.89 | 0.81 |
| | 2nd Winter | 0.95 | 0.96 | 0.93 |
| | 3rd Spring | 0.85 | 0.88 | 0.78 |
| | 3rd Winter | 0.95 | 0.95 | 0.93 |
| Testing | 1st Spring | **0.82** | 0.81 | 0.79 |
| | 1st Winter | **0.91** | 0.90 | **0.91** |
| | 2nd Spring | **0.77** | 0.76 | 0.74 |
| | 2nd Winter | **0.66** | 0.65 | 0.65 |
| | 3rd Spring | **0.69** | 0.68 | 0.67 |
| | 3rd Winter | **0.83** | 0.82 | 0.82 |

**Table 3:** Predictive accuracy for adding the predictors from P1 back to the best models of RF and XGB. Note: the best models of RF and XGB is selected based on the lowest RMSE on the testing dataset.

| | Basin | RF with P1 | RF without P1 | XGB with P1 | XGB without P1 |
|---|---|---|---|---|---|
| RMSE | 1st | **2.42** | 2.44 | **2.52** | 2.55 |
| | 2nd | **3.16** | 3.17 | **3.16** | 3.22 |
| | 3rd | **5.81** | 5.81 | **5.73** | 5.87 |
| Adj_ $R^2$ | 1st | **0.81** | 0.81 | **0.80** | 0.79 |
| | 2nd | **0.77** | 0.76 | **0.75** | 0.74 |
| | 3rd | **0.69** | 0.69 | **0.70** | 0.68 |