# Peer review of "Development of a Wilks Feature Importance Method with Improved Variable Rankings for Supporting Hydrological Inference and Modelling"

_Hydrology and Earth System Sciences, 2021_

## Author Response (AR1)

We would like to thank two anonymous referees for their time and valuable inputs to our manuscript. Their constructive suggestions and valuable comments are critical to improving the overall quality of the manuscript. Please find below a list of our responses to the comments and changes to the paper carried out to carefully address the remarks and the suggestions raised by the two referees. The comments are shown below in *black italic* font. The responses are shown below in blue.

**RESPONSES TO ANONYMOUS REFEREE #1'S COMMENTS**

*The authors developed an interesting approach of quantifying the relative importance of predictors for decision trees. Overall, the manuscript is well-organized, and the method and analysis appear to be well developed. To my knowledge, the interpretable machine learning has been receiving increasing attentions from hydrological community, and the proposed method would contribute to such a growing body of research. I have some concerns as indicated in the comments to the authors.*

We much appreciate the reviewer's positive comments, and have made efforts to greatly improve the manuscript by addressing the comments point by point.

*General comments:*
*The new method proposed in this study seems to quantify the importance of predictors from the training datasets and then validate the importance over the testing datasets through two evaluation metrics. The authors argue that the proposed method can identify key predictors that may be overlooked by other interpretation methods. By testing this hypothesis, those "overlooked" predictors are reconsidered into models, and this leads to improved accuracy on the testing dataset. The question is that the authors only tested that conventional interpretation methods may overlook some predictors but did not test whether there exist same issues regarding the proposed method. For instance, in Figure 7, the winter irrigation period of the 3rd irrigated watersheds, I7 was considered as important predictors by other models but not by the proposed method. So how will the model perform without this predictor? Similar concerns also exist for the other irrigated watersheds and irrigation seasons. Please list more evidence that the proposed methods can select more informative predictors.*

We would like to thank the reviewer for his/her insightful comments. To address the credibility of the proposed method, we carried out monthly streamflow simulations for 673 basins in the United States. The newly added simulations further supported our previous findings that the proposed WFI can provide more robust variable rankings than other methods. This was evidenced that WFI selected predictors help the random forest model achieve optimum predictive accuracy (refer to section 4 in the manuscript for details). We conclude that the WFI could produce more robust variable rankings, which enables a universal solution rather than a specific one for hydrological inference.

*Make it clear that what datasets are used for training each of the interpretation methods. I have seen many literatures using testing dataset to evaluate the feature importance, which I think is not appropriate. In this study, the authors seem used different datasets (training and validation) to calculate the importance scores under different interpretation methods. This*

*should be clarified. In addition, the validation datasets used in this study were not described clearly.*

We have added some related works describing different feature importance measures and clarifying the associated datasets used as suggested. The mean decrease impurity and the proposed WFI methods evaluate the importance scores during the tree deduction process, thus is based on training data. Permutation feature importance is a post-hoc evaluation measure (i.e., evaluate the importance score after the model is built), therefore, it is based on the out-of-bag (i.e., OOB) dataset. The training dataset is obtained from sampling from the original training set with replacement (the sample size is as same as the original training set). This process can leave about 1/3 of the training dataset as out-of-bag (OOB) data; these OOB data will not be involved in the model training process, and can thus be used for validation purpose. We clarified the OOB data under section 2.1.

*There are many small errors throughout the manuscript and need to be corrected. For example, authors sometimes use "irrigated watersheds" and sometimes use "drainage basins", similar phrases should be unified.*

We much appreciate the reviewer's careful review and have corrected inappropriate expressions to improve the manuscript's readability.

*Specific line comments:*
*Lines 20-21: make it clear that the predictors identified by WFI using the training dataset achieved the highest predictive accuracy on the testing dataset.*

We have replaced this sentence with a clearer one.

*Lines 24-25: the related findings should be more specific.*

We have rewritten the related findings as suggested: "…By employing the recursive feature elimination approach, our results indicated that the WFI could generate stable variable rankings in response to the reduction of irrelevant predictors. In addition, the WFI selected predictors can help RF achieve its optimum predictive accuracy, which indicates the proposed WFI could identify more informative predictors than other feature importance measures."

*Line 87: the regression tree ensembles are not necessarily composed of hundreds of interpretable models (i.e., decision trees). It is suggested to add the word "usually" before "composed" for clarity.*

We have revised the sentence as suggested.

*Lines 117-119: The meaning of this sentence is too vague, please clarify.*

In this revised manuscript, we removed the Bayesian model averaging (BMA) approach and only focused our study on WFI. Therefore, the associated descriptions for BMA (including this sentence) are removed.

*Line 129: Clarify the timesteps that used for model prediction. I believe the feature importance is evaluated based on the daily streamflow prediction, please clarify.*

We have clarified the timesteps of the two case studies in this research as suggested. Lines 160 to 164: "…Comparative assessment of WFI, PFI and MDI performances under the RF framework will then be undertaken through efforts in simulating monthly streamflows for 673 basins in the United States. With a finer temporal resolution, the proposed approach has also been applied to three irrigated watersheds in the Yellow River Basin, China, through concrete simulations for their daily streamflows."

*Line 133: "Yellow River Basin in China." Add a clarification that this is in China.*

We have added the clarification as suggested.

*Line 147: please add the word "daily" before "streamflow".*

We have added the clarification as suggested.

*Line 171: the abbreviate "MDI" shows before its definition in lines 177-178.*

We have added the abbreviate of "MDI" to the place where it first appears.

*Line 179: in Equation 3, "MDA" should be "MDI".*

This has been corrected.

*Line 184: the word "treatment" is not appropriate here.*

We have replaced the word "treatment" with "mechanism". Hence, the sentence becomes, "Such a mechanism naturally assumes that the predictors considered (for node splitting) in lower levels of the tree are less significant than those in upper levels."

*Line 261: Is the regression tree ensemble used in this study based on the random forest? Since there exist other regression tree ensemble approaches such as the extreme gradient boosting, this point should be clearly stated.*

We added the clarification in the manuscript that the stepwise cluster ensemble is based on random forest. Lines 285 to 290: "According to the law of large numbers, WFI is expected to perform better under the RF framework since the randomized predictors ensure enough tree diversity, which in turn, could lead to more balanced importance scores. We name the ensemble of SCA as the stepwise clustered ensemble (SCE). In addition to the three hyperparameters (i.e., *Ntree*, *Nmin* and *Mtry*) for Breiman's RF, SCE also requires significance level ($\alpha$), which is used for the $F$-test during the node splitting process."

*Lines 300-301: please clarify the phrase "effects from varied predictor characteristics".*

We have removed this expression since it is related to BMA.

*Lines 324-325: suggested revision: "daily streamflow for Spring-Summer (April to September) and Autumn-Winter (October to March) were modelled separately".*

We have revised the sentence as suggested.

*Lines 319-321: authors only mentioned training and testing datasets here. The "validation" period is not mentioned until the lines 336-337. Please clarify the model validation datasets at somewhere appropriate.*

We have added the validation (OOB) dataset under section 2.1. Lines 170 to 174: "The training set for building each tree is drawn randomly from the original training dataset with replacement. Such bootstrap sampling process will leave about 1/3 of the training dataset as out-of-bag (OOB) data, which thus can be used as a validation dataset for the corresponding tree."

*Lines 337-339: authors used two benchmark models as random forest and extreme gradient boosting. However, there were no descriptions for these two models throughout the paper. Please check.*

We have added random forest as related works in section 2.1 and removed the extreme gradient boosting (XGB), because we want to focus our study on comparisons among WFI, PFI and MDI, while XGB does not equip any of these feature importance measures as its default setting.

*Line 346: same issue with lines 319-321.*

We have added the validation (OOB) dataset under section 2.1.

*Line 354: please check if the MDI can be applied to the XGB model?*

Indeed, MDI is not an option for XGB model, so we removed XGB from this study.

*Lines 420-422: please explain why the less overfitted SCE-WFI can provide more informative predictors than others.*

We added a discussion part explaining the potential reasons why WFI may produce more robust variable rankings than MDI and PFI. In short, WFI evaluation process depends on Wilk's Λ, which prevent any splitting that due to chance. In the node splitting process, a predictor that significantly increases the predictive accuracy may not necessarily have the ability to differentiate two potential sub-spaces. Therefore, the WFI method (which evaluates every splitting and merging action based on Wilk's test-statistics with the predefined significance level α) is expected to generate more robust variable rankings. Second, WFI considers all the interactions among predictors in the tree deduction process, while PFI can only consider the

effect of one predictor at a time, thus the interactions between the target predictor and the rest predictors are overlooked.

*Line 430 (Figure 5): It is not clear what does letters A to E represent in this Figure. It is recommended to add some explanation in the figure or add notes.*

We have improved the figure presentation to avoid unclear expression. Please refer to Figure 13 in the revised manuscript for details.

*Line 452: what does the best-performance model indicate here? Does it refer to the model with smallest RMSE over the testing dataset?*

We chose the models with the smallest RMSE (among all the RFE iterations) on the testing dataset for the hydrological inference. To investigate whether the predictors identified by WFI are also meaningful to other algorithms, we reinserted the predictors in P1 into the best RF model (in which the set of predictors reaches the smallest RMSE over the testing dataset).

*Lines 468-469: please define the "accuracy-based interpretation methods".*

We have revised the expression as "feature importance measures relying on performance measures".

*Line 474: "hydrological processes" is a vague expression here, consider removing this phrase.*

We have removed this expression.

*Lines 516-517: how does the comparison lead to the statement?*

We have rewritten the discussion section and replaced this vague statement with WFI advantages. Please refer to section 6 (Discussion) for detailed revisions.

*Lines 733-735: clarify that the simulation period is Spring irrigation in Table 3.*

We have added the clarification as suggested.

**RESPONSES TO ANONYMOUS REFEREE #2'S COMMENTS**

*General comments*
*Overall, I believe that the paper is meaningful and interesting. Also, much work has been done for it. However, I currently have several major comments that, to my view, should necessarily be addressed and, therefore, I recommend major revisions. The key direction in which revisions should be made is the following: As the paper introduces a new splitting rule (that is of general use), it should be written accordingly and not as a work entirely placed in a hydrological context. More precisely, its introduction section should start by presenting detailed information on existing feature importance methods and splitting rules (thereby introducing the reader to the study's background), continue by stating what the new splitting rule offers compared to the existing ones (in summary), and lastly state that three hydrological applications are conducted. Most of the works cited in this review do likewise. Also, these applications should not be treated as if they consisted some type of proof, but as examples illustrating how the new method could be adopted in hydrology. A more appropriate proof that the method performs well compared to existing ones (e.g., an extended empirical investigation using a much larger dataset or a rigorous theoretical explanation) should also be provided, to my view.*

We much appreciate the reviewer for his/her constructive comments. To address these comments, we have substantially revised the introduction section as the reviewer suggested. We also added a new section (section 4), in which monthly streamflows for 673 basins in the United States were simulated to support our previous findings.

*Specific comments*
*1) To my view, the paper primarily focuses on a problem that is not hydrological −by its own nature− but algorithmic and more general, and it only secondarily presents a hydrological application. Therefore, it should be written accordingly and not as a work entirely placed in a hydrological context. In fact, the current version of the manuscript could confuse the reader, as it leaves the impression that the new method is motivated by hydrological discussions (mostly those made around the equifinality principle), and not by the machine learning literature and the need to provide better feature importance methods (which are, of course, needed in hydrology, but not only in hydrology).*

We much agree with the reviewer's suggestion and have changed our primary focus on the proposed WFI method rather than a hydrological problem. The WFI method is then used for supporting hydrological inference and modelling. We have rewritten the introduction section, which is framed as follows: it will start with (1) providing evidence that interpretable machine learning has been receiving increasing attention from the hydrological community, followed by (2) literature reviews on existing feature importance methods in general, feature importance methods for tree-structured models, recent development for splitting rules, and research gaps, after that (3) the proposed WFI method along with the splitting rules will be mentioned in summary and state that how does the proposed one compare with the existing ones, and finally (4) two real-world hydrological applications (monthly streamflow simulations for 673 basins in US and daily streamflow simulations for 3 irrigated watersheds in China) will be introduced.

*2) The motivation of the paper seems to be related to the equifinality principle-concept (which is extensively studied in hydrology). Nevertheless, it is unclear to me how the provision of better feature importance scores could solve the "equifinality problem". Further, I am not sure if equifinality could be referred to as a "problem", as it only implies that different modeling solutions could lead to outcomes of similar quality-value.*

We fully agree with the reviewer's suggestion. We have removed the equifinality principle from our introduction to clarify our motivation for this study (i.e., development of the WFI method that can be used to support hydrological inference and modelling).

*3) More generally, the concepts of equifinality, interpretability, collinearity and predictivity are explained, discussed and presented to be connected in a way that could confuse the reader. Thus, I think that the related background should be re-examined and that the related parts of the paper should be updated accordingly.*

We regret the unclear definition of some concepts (e.g., equifinality, interpretability, collinearity and predictivity). In the revised manuscript, we avoided using the concepts of equifinality, collinearity and predictivity. To avoid the confusion, we added the clarification to interpretability and stability in the manuscript as follows:

Lines 46 to 47: Interpretability can be defined as the degree to which a human can understand the cause of a decision (Miller, 2019).

Lines 121 to 122: Yu (2013) defined that statistical stability holds if statistical conclusions are robust or stable to appropriate perturbations.

*4) Key literature pieces on splitting rules and decision trees are also missing from the manuscript, despite the fact that they are necessary for covering the work's background.*

We have substantially revised the introduction and added a lot of literature for splitting rules and decision trees. Please refer to the introduction section from lines 83 to 146.

*5) Much effort has been put for placing this research into a hydrological framework. However, my general feeling is that there is something artificial about this, which does not even offer much to the paper, as it does not make the algorithmic-machine learning part smoother and easier to capture. To my view and given the main contribution of the paper (which is the introduction of a new feature importance method and a new splitting rule), more attention should be placed to which the existing feature importance methods are (e.g., the Gini, permutation, conditional permutation methods), which their theoretical properties are and what the new method does compared to them.*

We agree with the reviewer's suggestion. In the revised manuscript, we have focused on our main contribution as developing the WFI method as suggested. In detail, we have reframed our paper by adding a section of related works (section 2) describing random forest, permutation

feature importance and Gini importance (i.e., MDI) with detailed calculation processes. Then, WFI was presented in section 3 with comparisons of the existing splitting rule for RF.

*6) To my view, lines 249-258 are the most important part of the manuscript (that ideally should be extended and written rigorously), as they communicate some key advantages of the proposed method. However, it seems to me that these lines present only thoughts (and perhaps the overall rationale behind the method's conceptualization), and that they do not provide any proof, neither are they connected with other concepts discussed in the manuscript (e.g., equifinality, interpretability and more). Further, I am not sure that a real connection with these concepts exists.*

We have rewritten this paragraph as suggested. We demonstrate potential reasons why the WFI can lead to less biased variable rankings than MDI by using the synthetic case in Figure 1. Then in the next section, we used 673 basins in the United States as an empirical test to support our hypothesis.

The paragraph was revised as follows:

"There could be two potential advantages of WFI over MDI. First, the decrease in node impurity (DI) will become smaller and smaller as long as the tree level goes down (as shown in the bottom-right table in Figure 1). Such a mechanism naturally assumes that the predictors considered (for node splitting) in lower levels of the tree are less significant than those in upper levels. This effect is even aggravated by the existence of predictor dependence, which will depress the importance scores of independent predictors and increase the positively dependent ones (Scornet, 2020). As a consequence, some critical predictors may only receive small importance scores. In comparison, Wilk's $\Lambda$ is a measure of the separateness of two subspaces, which could avoid the above-mentioned issue for MDI because values of (1-$\Lambda$) do not necessarily decline as long as the tree level goes down (as shown in the bottom-right table in Figure 1). Therefore, the predictors that are primarily considered in latter splits still possible to own higher importance scores than those in early splits. As a consequence, some critical predictors might be identified by WFI but overlooked by MDI. Second, the node splitting mechanism of WFI is based on $F$-test, which, therefore, may significantly reduce the probabilities that the two child-nodes are split due to chance. Such a mechanism could be helpful to build more robust input-output relationships for prediction and inference by reducing overfitting. The above-mentioned potential advantages of WFI will be tested with a large number of hydrological simulations in the following two sections."

*7) More generally, the paper does not seem to follow one of the standard paths appearing in the literature for justifying the introduction of new algorithms in general and splitting rules for decision trees in particular. These paths are (a) the study of the method's asymptotic properties (i.e., an assessment through a theoretical investigation), and (b) empirical tests using large datasets (preferred when a theoretical investigation is too difficult). The three empirical examples currently presented in the paper do not provide either a theoretical or an empirical justification that the new method is well-designed (and, therefore, this justification is currently missing from the manuscript). Please note here that I do not mean to imply that the proposed method is worse than others. Instead, I think that further justifications are required at the*

*moment for the presentation of the new method to become complete and for its properties to be understood.*

We have extensively revised our manuscript and added monthly streamflow simulations for 673 basins in the United States as an empirical test. These basins across the contiguous United States that span a very wide range of hydroclimatic conditions with relatively low anthropogenic impacts (Newman et al., 2015). For each of these basins, recursive feature elimination (RFE) (Guyon et al., 2002) was carried out as follows: (1) train SCE and RF models with all predictors; (2) calculate the importance scores using the three interpretation methods embedded in their corresponding models; (3) exclude three least relevant predictors for each set of the importance scores obtained in step 2; (4) retrain the models using the remaining predictors in step 3; (5) repeat step 2 to 4 until the number of predictors less or equals to a threshold (set to 4 in this case study).

To directly compare the quality of variable rankings from different feature importance measures, the selected predictors from WFI (after every RFE iteration) were also used to train RF. This procedure allows the effects of different variable rankings to be solely from feature importance methods (i.e., removed the effects from different node splitting algorithms). The same procedure was also performed for SCE-based PFI (i.e., SCE-PFI) to examine whether the differences in variable rankings are from the WFI method or the tree deduction process in SCE.

*8) Furthermore, it is unclear to me (if and) how the Bayesian model averaging method supports in a straightforward way the assessment of the new feature importance method. Also, XGboost does not seem to be (absolutely) necessary for reaching the paper's objectives (as the boosting and random forest algorithms differ a lot). A general feeling of mine (which might be due to missing justifications) is that some of the methodological pieces are only artificially connected with the rest.*

Indeed, Bayesian model averaging (BMA) used in this study is a post-analysis approach to the proposed WFI method. BMA further investigates how the importance scores obtained by WFI vary in response to the variations of streamflow. We agree that such post-analysis is not necessary for this study and may even dilute the goal of this study; therefore, in the revised manuscript, we have removed this part from the method. We also agree that XGboost is not necessary for this research as the reviewer said, "boosting and random forest algorithms differ a lot", so we removed XGboost from the comparative studies.

*9) Additionally, some contradictory statements exist throughout the manuscript making the latter a bit hard to follow. For instance, in the abstract it is written that "the WFI has an advantage over PFI and MDI as it does not account for predictive accuracy so the risk of overfitting will be greatly reduced", while later it is written that "the comparative study also shows that the predictors identified by WFI achieved the highest predictive accuracy on the testing dataset".*

We are sorry for the confusion raised by the contradictory statements. In fact, WFI does not rely on performance measures (e.g., least-square errors in MDI or mean square errors in PFI) to evaluate the variable importance (over the training period). Instead, it depends on Wilk's $\Lambda$,

which prevent any splitting that due to chance. The improved variable rankings can better identify valuable predictors and eliminate predictors with a negative effect on predictive accuracy, so that the selected predictors may help achieve the optimum predictive accuracy over the testing period. In the revised manuscript, we have rewritten the statements as: "WFI does not rely on performance measures to evaluate the variable importance. Instead, it depends on Wilk's $\Lambda$, which prevent any splitting that due to chance."

*10) Papers that could be consulted for improving the presentation of the new method, in line with the above-provided comments, are listed here below (but many more exist, while an attentive literature review is currently missing from the paper):*
*a) Athey et al. (2019): This paper presents a splitting rule for maximizing heterogeneity.*
*b) Bénard et al. (2021): This paper provides better definitions and explanations of concepts like interpretability, simplicity and predictivity, and could help in putting the manuscript in a broader context, in the machine learning community.*
*c) Du et al. (2021): This paper presents another splitting rule.*
*d) Epifanio (2017): This paper presents an assessment of a new approach to assessing variable importance.*
*e) Friedberg et al. (2020): This paper presents another splitting rule.*
*f) Gregorutti et al. (2017): This paper investigates the relationship between correlation and permutation importance measures.*
*g) Ishwaran et al. (2008): This paper presents another splitting rule.*
*h) Roy and Larocque (2012): This paper presents another splitting rule.*
*i) Scornet (2020): This paper presents a theoretical investigation of the MDI. Also, it provides a clear discussion of the concept of interpretability.*
*j) Strobl et al. (2008): This paper presents the conditional variable importance metric, which is among the most popular and old variable importance metrics.*
*k) Wager and Athey (2018): This paper presents another splitting rule. It also introduces causality for random forests.*

We are very grateful to the reviewer for his/her careful and insightful reviews, which are very valuable for us to improve the presentation of the proposed method. We reviewed and cited lots of related papers including all the suggested ones. The detailed revisions can be found in the introduction section from lines 83 to 146. In summary, we first reviewed several splitting rules and related RF variants, followed by the reviews of RF-based feature importance measures and the research gap.

---

## Author Response (AR2)

We much appreciate the anonymous reviewers' valuable comments and suggestions. We have made revisions based on these comments and suggestions. We have also carefully proofread our manuscript to further improve its quality. Please find below our responses to each of your comments.

**Comment from Anonymous reviewer 1:**

Most of my comments for last version have been well clarified. I only have one minor comment:

The results in Table 2 is a little bit weird. The performance of SCE is always 0.01 higher in adjusted R^2 than RF. Please double check them.

**Response:**

We have double-checked our results, and they are correct. To dissolve such concerns from the potential audience, we kept three digits instead of two, using percentages for improved clarification.

**Revisions:**

**Table 2**: The adjusted $R^2$ for SCE and RF with all considered predictors.

| Basin | Season | Training | | OOB | | Testing | |
|---|---|---|---|---|---|---|---|
| | | SCE | RF | SCE | RF | SCE | RF |
| 1st | spring | 94.1% | 98.2% | 86.5% | 88.2% | **82.0%** | 81.4% |
| 1st | winter | 97.9% | 99.2% | 94.3% | 95.1% | **91.3%** | 90.0% |
| 2nd | spring | 94.0% | 98.4% | 85.7% | 89.0% | **76.7%** | 75.7% |
| 2nd | winter | 97.6% | 99.3% | 94.6% | 95.7% | **66.0%** | 65.1% |
| 3rd | spring | 93.8% | 98.3% | 84.5% | 87.7% | **68.5%** | 68.1% |
| 3rd | winter | 97.8% | 99.2% | 95.2% | 95.3% | **82.7%** | 82.1% |

**Comment from Anonymous reviewer 2:**

My comments have been fully addressed.

**Response:**

Thank you again for your time and efforts. Much appreciated.